# Microfluidic Chip for Detection of Drug Resistance at the Single-cell Level

**DOI:** 10.3390/mi14010046

**Published:** 2022-12-25

**Authors:** Kena Song, Zhangqing Yu, Xiangyang Zu, Lei Huang, Dongliao Fu, Jingru Yao, Zhigang Hu, Yun Xue

**Affiliations:** 1College of Medical Technology and Engineering, Henan University of Science and Technology, Luoyang 471023, China; 2Chongqing Key Laboratory of Soft Condensed Matter Physics and Smart Materials, College of Physics, Chongqing University, Chongqing 401331, China

**Keywords:** antimicrobial susceptibility testing (AST), single-cell level, drug prepositioning, microfluidic chip

## Abstract

Drug-resistant bacterial strains seriously threaten human health. Rapid screening of antibiotics is urgently required to improve clinical treatment. Conventional methods of antimicrobial susceptibility testing rely on turbidimetry that is evident only after several days of incubation. The lengthy time of the assay can delay clinical treatment. Here, we proposed a single-cell level rapid system based on a microfluidic chip. The detection period of 30 min to 2 h was significantly shorter than the conventional turbidity-based method. To promote detection efficiency, 16 independent channels were designed, permitting the simultaneous screening of 16 drugs in the microfluidic chip. Prepositioning of drugs in the chip permitted prolonged transportation and storage. This may allow for the widespread use of the novel system, particularly in the regions where medical facilities are scarce. The growth curves were reported rapidly through a custom code in Matlab after tracking and photographing the bacteria during microscopy examination. The capability of the proposed system was validated by antimicrobial susceptibility testing trials with standard strains. The system provides a potentially useful detection tool for drug-resistant bacteria.

## 1. Introduction

Multi-antibiotic-resistant bacterial infections are a stated concern of the World Health Organization [1]. Antibiotic-resistant bacteria are an acknowledged and increasingly prevalent problem in hospitals. These include methicillin-resistant *Staphylococcus aureus* and vancomycin-resistant Enterococci. Antibiotic resistant bacteria have also been detected in the residential environment. A discovery and developmental pipeline of new antibiotics is needed to combat bacterial antibiotic resistance [2,3,4]. Antimicrobial suscept ibility testing (AST) is essential to identify the proper antibiotic for a given bacterial infection. However, the flood of materials with potential antibiotic activity adds to the burden of drug screening [5,6]. 

Traditional bacterial resistance analyses, such as the Kirby-Bauer paper method (K-B method) and agar dilution, are burdened by the long incubation time required [7,8,9,10]. These detections rely on plaque production or the turbidity of the bacterial solution; both events develop during several days of incubation. This long period adds to the medical burden and risks delaying treatment for patients. In recent years, polymerase chain reaction (PCR) and quantitative PCR detection have been applied in analyses of bacterial resistance, by detecting drug resistance related genes [11,12,13,14,15,16]. Although drug resistance genes can be detected rapidly and accurately, the PCR-based technology can only be used to screen known target genes and cannot be used in large-scale and high-throughput methods [17,18]. In contrast, the combination of whole genome sequencing technology and matrix-assisted laser desorption/ionization-time of flight mass spectrometry (MALDI-TOF MS) detection technology could discover as many drug resistance mechanisms as possible and rapidly screen for sensitive drugs [19,20,21,22]. As a valuable means of studying the mechanisms of bacterial drug resistance, MALDI-TOF MS is only applicable to relatively few bacteria, based on the principle of bacterial drug resistance [23]. Therefore, it is urgent to seek a rapid, efficient detection platform of bacterial resistance. 

The single-cell level methodology based on a microfluidic chip provides an opportunity to realize the rapid detection of bacterial resistance [24,25]. It does not require prolonged incubation to allow for the development of turbidity or plaques. Rather, the proliferation rate is determined in a novel manner by counting cell number accurately in a short time. Furthermore, the approach is able to quantify the accumulation of antibiotics in bacteria [26]. Here, we propose a rapid AST system with single-cell resolution. The system combines microfluidic chip technology and an automatic image recognition system [27,28,29,30]. The bacterial state is tracked by enumerating cells instead of bacterial plaques or turbidity, which shortens the detection period from 30 min to approximately 2 h. The prepositioning of the drug in the microfluidic chip allows for prolonged transportation and storage. The system could have widespread applications, especially in regions with few medical facilities. The bacteria in the microfluidic chip are incubated in a living cell device that is positioned on the stage of a microscope. A suitable environment is provided in the confined space, including temperature, gases, and humidity. The appropriate incubation environment is conducive to the real-time monitoring of bacteria. The images are synchronously captured during incubation by microscopy and processed by the automatic image recognition technology. To reduce the platform barriers, simple bright-field photos were adopted. In order to retain more edge information while smoothing the noise, the traditional Canny edge algorithm is improved by bilateral filtering instead of gaussian filtering to segment the high noise images. The convolution weight of bilateral filtering not only depends on the position relation of the pixel spatial domain, but also considers the similarity degree between pixels. After edge segmentation, binarization, and counting, the growth curve is reported. The process is shown in the top portion of Figure 1, with the devices and instruments used displayed in the bottom portion. The system was validated by AST trials with the standard strains of *E coli* (Top10, pEGFP-N) and *E. coli* (Top10, pET32a). 

## 2. Experimental Methods

### 2.1. Chip Design and Manufacturing

Three important zones are designed in the microfluidic chip: drug-injection, bacterial-injection, and reaction zones. The reaction zone is also the observation zone in the later stage. The overall structure of the chip is shown in Figure 2a. Sixteen flow channels (indicated in red) are isolated from the bacterial-injection well. These allow the antimicrobial susceptibility of 16 drugs to be independently tested. Correspondingly, 16 drug channels (indicated in green) are designed in opposition, which are used for drug delivery. The drugs are injected from the drug-injection well. The reaction zone is designed in the junction of the two kinds of channels, i.e., circle chambers (indicated in orange) located in the center of the chip. To facilitate bacterial counting, the depth of the chambers is approximately 15 µm. The volume capacity of the bacterial suspension is limited by the geometric depth of the reaction zone, which combines the diluted bacterial suspension and the reserved settlement time to ensure the bacteria are retained as a single layer in the chamber.

The microfluidic chip is fabricated as a polydimethylsiloxane (PDMS) layer with patterns and glass as the substrate. The five-step (I–V) fabrication process is shown in Figure 2b. In step (I), lithography is used to generate the mold. In step (II), 5 µL of release agent (trichloro (1H, 1H, 2H, 2H-perfluorooctyl) silane) is evaporated in the mold at 120 °C for 5 min to modify the surface. This modification is helpful to restrict PDMS peeling from the mold after curing. In step (III), the PDMS prepolymer Sylgard184 and the curing agent (DowCorning, Midland City, Michigan, USA) are mixed at a mass ratio of 10:1, mixed intensively, and the bubbles are removed. The mixed solution is poured on the prepared mold and cured at 60 °C for at least 4 h. In step (IV), the cured PDMS is peeled off from the mold and the injection wells are formed by drilling. In step (V) the PDMS layer is plasma-bonded to a glass substrate.

### 2.2. Bacteria Strains and Drugs

*E. coli* (Top10, pEGFP-N) and *E. coli* (Top10, pET32a) were used in the drug resistance testing. Both the strains were purchased from Takara Biomedical Technology (Beijing) Co., Ltd (San Jose, California, USA). The bacteria were incubated in a Luria-Bertani (LB) medium overnight to a high bacterial density. In the drug resistance testing, the suspension of bacteria was diluted to 1:10 and seeded in the chip. *E. coli* (Top10, pEGFP-N) detects resistance to Kanamycin and *E. coli* (Top10, pET32a) detects resistance to Ampicillin. Kanamycin and Ampicillin (Solarbio, Beijing, China) were chosen to verify the chip performance.

### 2.3. Drug Prepositioning and Use of the Microfluidic Chip

Drugs that had been lyophilized by the established technique of freeze-drying were prepositioned in the microfluidic chip. The freeze-drying technique was used for drug prepositioning. This is a low-temperature drying technique in which the solvent is removed under a freezing state through low-temperature sublimation with vacuum assistance. The freeze-drying technique is widely applied in the biomedicine field, since it avoids the high-temperature decomposition of drugs. The process of drug prepositioning is shown in steps I-V of Figure 3. For increasing the adhesion to the inner surface of the reaction chamber, poly-D-lysine (PDL) (Beyotime Company, Shanghai, China) was used as the surface modifier. After surface modification, the drug was quantitatively injected into the reaction zone using a microflow pump. Then, the chip was put into a lyophilizer (Shengchao Kechuang (Beijing) Biotechnology Company, Beijing, China) at −40 °C for 4 h until the drug was frozen, dehydrated, and had adhered to the surface of the reaction zone. The completed chip was vacuum-packaged for storage. The morphology after drug prepositioning was revealed through scanning electron microscopy (SEM) using a model JSM-7800F microscope (JEOL, Tokyo, Japan).

### 2.4. Drug Resistance Testing

The suspension of bacteria was injected as a single volume into the 16 reaction zones using the microflow pump. The drug rapidly dissolved in the medium. The chip was set into the miniature cell incubators positioned on the objective stage of a bright-field microscope for in situ culturing and observation. Proliferation of the bacteria was continuously tracked by taking photographs at 5 min intervals for 30 min~2 h. The photos were analyzed to enumerate the bacteria. The collective data were used to generate the growth curve automatically by Matlab. The test process is illustrated in panels VI–IX of Figure 3.

## 3. Results and Discussion

### 3.1. Verifying Effect of Drug Prepositioning by SEM

To reduce the operation steps, the drug is prepositioned in the microfluidic chip by vacuum freeze-drying. This permits the drug-containing chip to be stored for a long time and makes transportation to the site of testing convenient. These benefits are favorable for the use and scope of applications of the microfluidic chip. Figure 4 shows the microphotograph of the inner surface of a chamber after drug prepositioning. No signal could be captured before drug prepositioning (Figure 4a), but was after application of Ampicillin (Figure 4b) and Kanamycin (Figure 4c). 

### 3.2. Tracking Bacteria Growth in the Rapid AST System at the Single-cell Level

Bacteria are rapidly enumerated at the single-cell level in the AST system compared to traditional turbidity detection. The incubation period was shortened from several days to times ranging from 30 min to approximately 2 h. The bacteria were incubated in a living cell device providing suitable environmental conditions of 37 °C, 5% CO_2_, and hospitable humidity. After combining the constrained geometric space, dilution of the suspended bacteria, and the reserved settlement time, the bacteria grew as a single layer in the microchamber. This monolayer was convenient for enumeration. Figure 5a shows the images of bacteria in a time series in the microfluidic chip. The bacteria were clearly visualized by bright-field microscopy with a 40× objective lens (0.349 μm/pixel). The photograph shows a 475 μm field of view made up of 1360 × 1024 pixels. A representative image of the growth of *E. coli* within 40 min is displayed in Figure 5. During the short period of time, the numbers of bacteria significantly increased. The images were post-processed and counted to construct a growth curve, such as that shown in Figure 5c. The growth rate of 27.586% was computed simultaneously. 

### 3.3. Rapid Identification and Counting of Bacteria

In order to accurately enumerate bacteria from the microscopy images, an automatic bacterial counting plug-in was developed in the Matlab environment. The original microscopy images in TIF file format (Figure 5a) were imported into Matlab. Each image was displayed with the array of 1360 × 1024 in Matlab, representing the gray values (0–225) of the corresponding pixel. The traditional Canny edge algorithm was improved by bilateral filtering instead of gaussian filtering to extract the edges of high noise bright-field images. According to the distribution of pixels and edge segmentation algorithm, each image was converted into a binary image (Figure 5b). The connection regions were counted, which represents the numbers of bacteria. The improved Canny edge algorithm was verified repeatedly by comparison with counting by hand. The mean error of the automatic counting was approximately 0.03, indicating no interference with the statistics of proliferation tendency. Moreover, the multiple independent reaction zones provide the possibility to set several parallel tests to avoid false positives. Combining with the batch calculation method and the average of parallel tests, the bacterial growth curve was drawn to show the numbers of bacteria with time. The growth rate was calculated as a quantitative parameter. The improved Canny edge algorithm was used for image segmentation, specifically aiming at bacterial images with low recognition. A plug-in facilitated the communication with the image acquisition device to realize the automatic process from image acquisition to growth curve generation. Finally, the analysis results were displayed in a visual interface, as shown in Figure 5c. In the visual interface, two buttons of “file” and “rate of growth” are designed for convenient operation. These commands permit importation of the files and calculate the growth rate, respectively. The growth rate is displayed in the back box with red font. 

### 3.4. Validation of the Single-cell Level Rapid AST System with E. coli

To demonstrate the accuracy of the single-cell level rapid AST system, we evaluated the antibiotic susceptibilities of *E. coli* (Top10, pEGFP-N; Kanamycin resistance) and *E. coli* (Top10, pET32a; Ampicillin resistance) in the single-cell level rapid AST system. Kanamycin and Ampicillin were chosen in the verification test. The recommended concentrations of 50 and 100 µg/mL were matched and injected into the reaction chambers, then the drug was freeze-dried and adhered to the inner surface of the chambers. The bacteria suspension was injected into the chambers and was incubated in the living cell device positioned on the objective stage of the microscope. Before taking photographs, 30 min were allowed for drug dissolution and waiting for bacteria to settle and begin to proliferate. Photographs were taken every 5 min, continuing for 40 min to track drug susceptibility. 

In the test, obvious differences were revealed (Figure 6a,b). In the presence of Ampicillin, the growth rate of *E. coli* (Top10, pET32a) was 17.283%, which is consistent with the control group of 18.793%, which was within the margin of error. The pronounced resistance of *E. coli* (Top10, pET32a) to Ampicillin was rapidly verified at the single-cell level in the AST system. In the presence of Kanamycin, the growth rate of *E. coli* (Top10, pET32a) was just 0.861%, indicating pronounced sensitivity to Kanamycin. *E. coli* (Top10, pEGFP-N) was highly resistant to Kanamycin and highly sensitive to Ampicillin. The results are consistent with conventional detection methods. 

We also verified the effect of the drug prepositioning process from freeze-drying to dissolution, compared with the procedure in which the freeze-drying step was omitted. *E. coli* (Top10, pET32a) exposed to Ampicillin (Figure 6c, shown in red) and *E. coli* (Top10, pEGFP-N) exposed to Kanamycin (Figure 6d, shown in blue) were tested. The blue group denotes drugs added to the bacterial medium directly and at the same dose used in the presence of the bacteria. The results were consistent of the two groups within deviations. 

## 4. Conclusion

A single-cell level rapid AST system is proposed and verified based on a microfluidic chip and automatic image recognition technologies. The novel method is different from the traditional method of judging bacterial resistance by turbidity of bacterial suspensions or formation of bacterial plaque. Using microscopy image acquisition and automatic image recognition, bacteria are enumerated at the single-cell level, which shortens the required incubation time from several days for the traditional approach to only 30 min to approximately 2 h. The system is compatible with high-throughput simultaneous screening of up to 16 drugs on samples in the microfluidic chip. Compared with traditional manual cultivation and counting methods, real-time image acquisition could be completed, analyzed, and growth curves automatically generated quickly. The method’s speed and simple operation will benefit patient treatment. The abilities of long distance transportation and prolonged storage make this the novel method for widespread application. 

## Reference 

## Figures and Tables

**Figure 1 micromachines-14-00046-f001:**
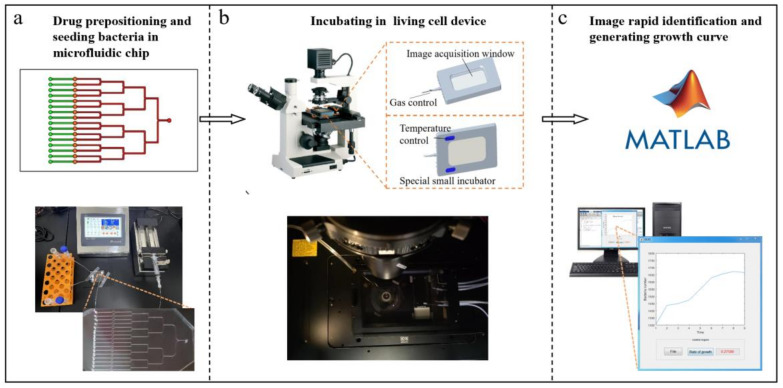
Process of the single-cell level rapid AST system. (**a**) Bacteria are injected into the microfluidic chip using a microflow pump. (**b**) The microfluidic chip is placed in the living cell device positioned on the microscope stage for incubation and real-time detection. (**c**) Photos captured by the microscope are processed by the Matlab software to generate the growth curve. The hardware used are also displayed.

**Figure 2 micromachines-14-00046-f002:**
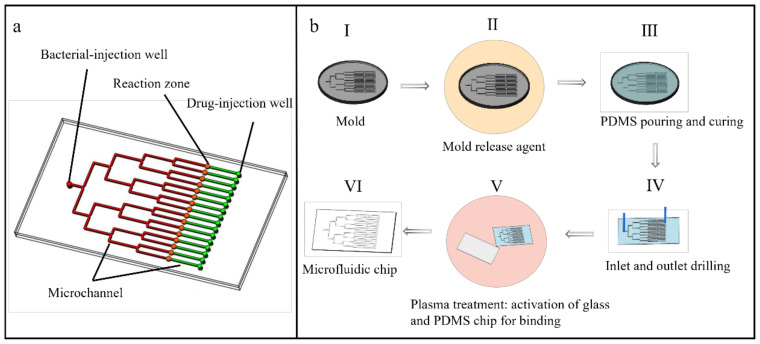
Structure and fabrication process of the microfluidic chip. (**a**) Diagram of the chip structure. The microfluidic chip contains three important zones: bacteria injection zone (red), drug-injection zone (green), and reaction zone (orange). (**b**) Fabrication of the microfluidic chip. (I) Manufacture of the chip mold on silicon wafer by photolithography. (II) Fumigation of release agent on the mold. (III) Pouring of PDMS mixed with curing agent onto the mold, followed by curing in a drying oven. (IV) Peeling the solidified PDMS from the mold and punching at the injection region. (V) Encapsulating the microfluidic chip by plasma bonding. Panel VI displays the completed chip.

**Figure 3 micromachines-14-00046-f003:**
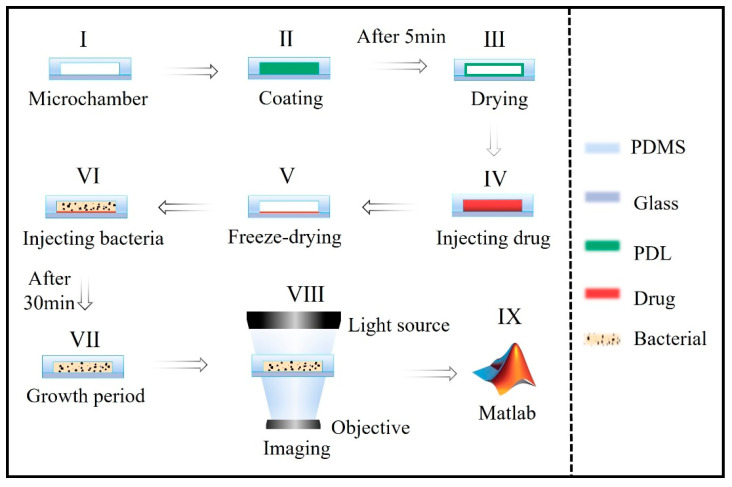
The operational process of the single-cell level rapid AST system. (I) Side view of the section of the micro chamber. (II) Injection of PDL to modify the microchamber. (III) Microchamber after coating with PDL. (IV) Injection of drug in the microchamber. (V) Freeze-drying of drug using the lyophilizer. (VI) Injection of bacterial suspension into the microchamber. (VII) The freeze-dried drug dissolves in the bacterial suspension and time is allowed for cell proliferation. (VIII) Tracking bacteria proliferation by microscopy with photographs taken every 5 min as one photo. (IX) Enumeration of bacteria at each time point and Matlab automatic compilation of data and generation of growth curve.

**Figure 4 micromachines-14-00046-f004:**
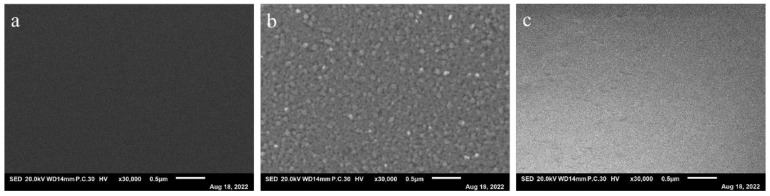
SEM images of the inner surface of microchamber with 30000× magnification. (**a**) Before drug prepositioning. (**b**) After Ampicillin prepositioning by freeze-drying. (**c**) After Kanamycin prepositioning by freeze-drying.

**Figure 5 micromachines-14-00046-f005:**
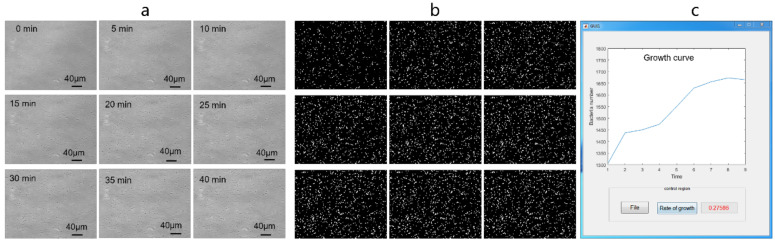
Analysis process of the automatic bacterial counting plug-in. (**a**) Photographs of bright-field microscopy images captured at defined times using a 40× objective lens. The images were the original files imported into the plug-in. (**b**) Conversion of the bright-field photographs to binary images. (**c**) Growth curve generated in the visual interface.

**Figure 6 micromachines-14-00046-f006:**
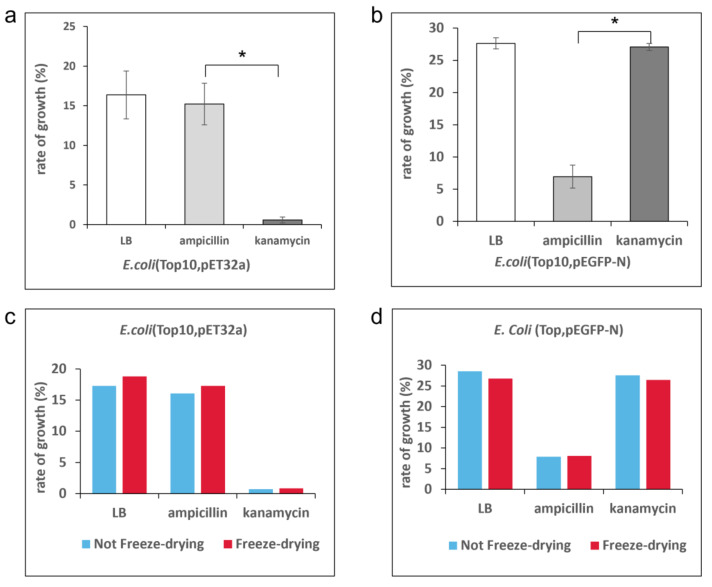
The validation test with *E. coli* in the single-cell level rapid AST system. (**a**) Growth rates of *E. coli* (Top10, pET32a) in the presence of Ampicillin and Kanamycin. *E. coli* (Top10, pET32a) shows high sensitivity to Kanamycin and high resistance to Ampicillin. (**b**) Growth rate of *E. coli* (Top10, pEGFP-N) in the presence of Ampicillin and Kanamycin. *E. coli* (Top10, pET32a) shows high sensitivity to Ampicillin and high resistance to Kanamycin. LB is the blank control group of antibiotic-free medium. (**c**) Consistent results for the absence and presence of drug prepositioning for *E. coli* (Top10, pET32a) in the presence of Ampicillin and Kanamycin. (**d**) Consistent results of the absence or presence of drug prepositioning for *E. coli* (Top10, pEGFP-N) in the presence of Ampicillin and Kanamycin. (* *p* < 0.05.).

## Data Availability

Data available on request from the authors.

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
