# Peer review of "Microfluidic Chip for Detection of Drug Resistance at the Single-cell Level"

_micromachines, 2022, doi:10.3390/mi14010046_

Round 1
Reviewer 1 Report
This article proposes a single-cell level rapid AST system based on microfluidic chip which shortens detection period largely. 16 independent channels make it possible to promote efficiency and reflect the high-throughput feature of microfluidic chip. At the same time, it combines MATLAB to report the growth curve. In general, this article describes in detail how the system being established, but some details should be modified before being publication.
1. Because the real picture of microflow pump is used, I suggest the author should use the real pictures of the devices and instruments. At the same time, a schematic of whole procedure should be added in figure 1.
2. The figure c in figure 1 shows using MATLAB to report the growth curve which is too shallow to display, I suggest replace it with MATLAB interface or an example growth curve.
3. The bacterial solution was one injected into the 16 reaction zones by microflow pump. How to ensure the uniformity of sample injection? If not, will it affect the accuracy of experimental results.
4. Whether the cell count, batch effect and details of the canny algorithm ensure the accuracy of the experimental results or not? The reproducibility of experimental results cannot be guaranteed without original data and code.
5. There is no concluding caption in figure 2.
Reviewer 2 Report
§ In this manuscript, authors reported a microfluidic chip for rapid susceptibility testing with single-cell level detection. In general, the manuscript lacks the scientific and technical merits to warren publication. Also, the language is below the average of acceptable level. the manuscript English needs to be comprehensively revised and improved.
§ Authors need to elaborate on the rational of the study. Why single cell detection is important and advantageous comparing to turbidity detection?
§ The manuscript lacks the necessary technical discussion and the merits of the developed techniques. For example, there is no technical description of the optical system used in the image acquisition to capture the bacterial growth over time. What is the magnification, field of view, pixel size, etc. How the images interpreted by the Matlab program?
§ Freeze-drying methods need to be explained otherwise a reference can be cited.
§ “The images are synchronous captured in incubation by microscope and processed by intelligent image recognition technology.” What technology is used? This need to be mentioned clearly in the introduction.
§ The bacteria strains and drugs used in the study should be listed after section 2.1 as a separate sub section.
Minors
§ The title needs to be re-written in better way. Suggested: “Microfluidic chip for drug-resistant detection with single-cell level”.
§ This paragraph “Long-time incubation period is the limitation in the traditional bacterial resistance analysis, including Kirby Bauer paper method (K-B method), AGAR dilution method and broth dilution method, E-TEST strip method, even in the automated detection system of Meriere VITEK 2 automatic bacterial identification and drug sensitivity analysis system, Thermo automatic drug sensitivity analysis system, BD Phoenix Automated Microbiology system, etc.[7–10]” is misleading and needs to be rephrased.
§ “However, such technology could only be used to screen the known target genes, but not competent to detect and monitor bacterial resistance with large-scale and high-throughput.” But should be replaced with “Also, these methods are not ..”
§ The components in Fig.2a are not labelled according to the text in section 2.1.
§ “To facilitate bacteria counting, the depth of the chambers is designed just about 15 μm to ensure the bacteria retained single layer in the chamber.” how the channel with a depth of 15 μm ensures single bacteria retention?
§ What release agent was used to facilitate releasing the PDMS from the mold?
§ “Due to the otherness of drug characteristics, the surface morphology is difference.” Must be re-phrased
Reviewer 3 Report
The Authors have demonstrated a microfluidic chip for drug resistance detection of bacteria at single cell level.
The imaging and detection algorithm section does not describe about measures taken for false positives. Presence of that would be helpful and strengthen the paper.
Overall the paper is written in simple language and is easy to understand.
